# Deep Transfer Learning for Ni-Based Superalloys Microstructure Recognition on *γ*′ Phase

**DOI:** 10.3390/ma15124251

**Published:** 2022-06-15

**Authors:** Wenyi Li, Weifu Li, Zijun Qin, Liming Tan, Lan Huang, Feng Liu, Chi Xiao

**Affiliations:** 1College of Science, Huazhong Agricultural University, Wuhan 430070, China; wenyili@webmail.hzau.edu.cn (W.L.); liweifu@mail.hzau.edu.cn (W.L.); 2Hubei Key Laboratory of Applied Mathematics, Hubei University, Wuhan 430062, China; 3State Key Laboratory of Powder Metallurgy, Central South University, Changsha 410083, China; zijun.qin@csu.edu.cn (Z.Q.); limingtan@csu.edu.cn (L.T.); lhuang@csu.edu.cn (L.H.); liufeng@csu.edu.cn (F.L.); 4Key Laboratory of Biomedical Engineering of Hainan Province, School of Biomedical Engineering, Hainan University, Haikou 570228, China

**Keywords:** superalloys, scanning electron microscop, deep transfer learning, microstructure characterization, software, accelerating design

## Abstract

Ni-based superalloys are widely used to manufacture the critical hot-end components of aviation jet engines and various industrial gas turbines. The analysis of Ni-based superalloys microstructures is an important research task during the design and development of superalloys. The material microstructure information can only be understood by experts in the long history. Image segmentation and recognition are developing techniques for accelerating the microstructure analysis automatically. Although deep learning techniques have achieved satisfactory performance, they usually suffer from generalization, i.e., performing worse on a new dataset. In this paper, a deep transfer learning method which just needs a small number of labeled images is proposed to perform the microstructure recognition on γ′ phase. To evaluate the effectiveness, we homely prepare two Ni-based superalloys at temperatures 900 ${}^{{}^{\circ}}$C and 1000 ${}^{{}^{\circ}}$C, and manually annotate two datasets named as W-900 and W-1000. Experimental results demonstrate that the proposed method only needs 3 and 5 labeled images to achieve state-of-the-art segmentation accuracy during the transfer from W-900 to W-1000 and the transfer from W-1000 to W-900, while enjoying the advantage of fast convergence. In addition, a simple and effective software for the Ni-based superalloys microstructure recognition on γ′ phase is developed to improve the efficiency of materials experts, which will greatly facilitate the design of new Ni-base superalloys and even other multicomponent alloys.

## 1. Introduction

Ni-based superalloys [1,2] have been widely used in aerospace, marine, energy and other fields because of its high temperature strength, good oxidation resistance, corrosion resistance, fatigue resistance and other comprehensive properties, which are fundamentally controlled by the internal microstructure and compositions [3,4,5]. Ni-based superalloys usually contain more than eight metal elements. The modification on any alloy composition may cause a significant change of the precipitated phase and mechanical properties [6]. The phases of Ni-based superalloys generally include γ, γ′, TCP and GCP phases, where γ′ is the major phase yielding precipitation strengthening and its volume fraction and particle size play dominating roles in alloy performance [7,8]. Therefore, it is essential to obtain the accurate and reliable data on γ′ phase distribution, which can be achieved using microstructure recognition technology [9,10].

Typically, the evaluation and analysis of the microstructure are conducted by experienced experts with knowledge of materials and microstructural characterization [11,12,13]. This artificial process requires specific training and it is sluggish. With the development of computer science, microstructure can be recognized automatically. To be specific, the microstructure recognition [14] involves the image acquisition, processing, segmentation, information extraction, statistical analysis, and visualization. The subject of microstructure segmentation consists of a set of problems that are central to the disciplines of microstructure recognition.

Currently, a commonly used method for microstructure recognition of Ni-based superalloys is based on threshold [15]. However, it is difficult to find a fixed threshold to process different images or different regions in one image, especially when the image background or the feature intensities are heterogeneous. To this end, an adaptive method was developed and implemented to automatically adjust the threshold value for local pixel intensity [16]. Although the adaptive threshold seems to be a robust method to circumvent the above limitations, it also requires other advanced filtering methods and to be adjusted manually by materials experts. Recently, Kondo et al. [17] demonstrated that deep learning is useful for microstructure recognition to reveal the microstructure-property linkage in materials. Jia et al. [18] proposed an end-to-end network architecture that could accurately identify the γ′ phase in Ni-based superalloys. Although achieving satisfactory performance, these methods usually suffer from generalization when the data distribution changes. It means that existing methods will perform worse on a new Ni-based superalloys dataset produced at different conditions. In fact, they require a large number of pixel-level labeled images to train a new model, which is labor-intensive and time-consuming.

To address this problem, we propose to adopt deep transfer learning for microstructure recognition on γ′ phase due to its success in computer vision [19,20]. The objective is to settle how to use a small number of labeled images to achieve outstanding γ′ phase recognition for a new Ni-based superalloys dataset. The main contributions of this paper can be roughly grouped in three different directions summarized as follows:**Method:** We propose a deep transfer learning method for identifying the γ′ phase on Ni-based superalloys datasets. It can take full use of existing labeled images and train the deep learning model with a few labeled images. To the best of our knowledge, it may be the first endeavor on this topic;**Dataset:** To evaluate the effectiveness of the proposed method, two Ni-based superalloys are prepared at temperatures 900 ${}^{{}^{\circ}}$C and 1000 ${}^{{}^{\circ}}$C, and two datasets are annotated manually after acquisition and processing;**Application:** We develop a simple and effective software for the Ni-based superalloys microstructure recognition on γ′ phase, to help the materials experts. The code has been provided publicly (github.com/258yujin/transfer_learning_software, accessed on 23 April 2022) and can also be applied to other multicomponent alloys.

The rest of this paper is organized as follows. Section 2 introduces the high throughput experimental preparation process of Ni-based superalloys. Section 3 states the details of the proposed deep transfer learning method. Section 4 reports the experimental analysis of our approach. Section 5 presents the details of the developed software. Finally, Section 6 closes this paper with conclusion.

## 2. Materials

A high throughput method was applied to obtain adequate microstructural information. As a typical high throughput sample with gradient composition, multicomponent diffusion multiple (MCDM) was designed and employed to obtain a composition-dependent microstructure [21]. Taking W1-W3-W4 (designed as NiX-6W-6Mo) diffusion triple as an example to illustrate the steps for MCDM preparation and characterization, as shown in Figure 1a, the MCDM was obtained through assembling alloys with different compositions, as listed in Table 1, followed by electron-beam welding, hot isostatic pressing, and heat treatment. Thereafter, wide composition gradients and diverse microstructures were generated in a single sample. To observe the precipitates, the sample was etched in the solution of 33% H_2_O + 33% acetic acid + 33% HNO_3_ + 1%HF for 5–10 s, and characterized via Zeiss Supra 55 field emission scanning electron microscope (SEM) equipped with an ATLAS large-area imaging software, which could automatically capture and join images. To be specific, the SEM images with a pixel resolution of 20 nm were obtained by using back-scattered electron (BSE) and secondary electron (SE) modes simultaneously. As shown in Figure 1b, there is a big difference between the microstructures at temperatures 900 ${}^{{}^{\circ}}$C and 1000 ${}^{{}^{\circ}}$C. Thus, the integrated images with different precipitate characteristics were obtained for further microstructure recognition.

## 3. Methods

In this section, the deep transfer learning method for identifying the γ′ phase on Ni-based superalloys datasets is introduced. To better evaluate and illustrate the method, we have manually annotated the pixel-level γ′ phase from the SEM images of two Ni-based superalloys produced at temperatures 900 ${}^{{}^{\circ}}$C and 1000 ${}^{{}^{\circ}}$C, which are named as W-900 (including 148 images with size 512 × 512) and W-1000 (including 100 images with size 512 × 512), respectively. Given the whole labeled images in the W-900 and a few labeled images in the W-1000, the schematic flowchart of the proposed transfer method from W-900 (referred as source domain) to W-1000 (referred as target domain) is presented in Figure 2.

The training process of deep transfer learning is roughly summarized as follows. First, we train the basic network fully with the whole 148 images in the W-900. During the training, the Adam algorithm is adopted with a learning rate of 0.0001 to optimize the model. The batch size is set as 2 and the epoch is set as 50. Meanwhile, the data augmentation, including flipping, rotation and adding Gaussian noise is adopted to improve the generalization performance. It is well known that the shallow layers of deep neural network extract low-level general features that are suitable for common tasks, while the deep layers extract more advanced features that are highly dependent on specific data and mission [22]. Inspired by Ref. [22], we similarly freeze the weights in the encoder layers and fine-tune the weights in the decoder layers by a small number of images in the W-1000. During the fine-tuning, we adopt the common strategy of halving the initial learning rate in the Adam algorithm since the model has converged in the previous training [23]. It can guarantee that the final model has better generalization ability for the target domain.

In the proposed deep transfer learning method, we employ the U-Net encoder-decoder architecture as the basic network due to its remarkable simplicity, efficiency, and robustness [24]. In the encoder (feature extraction branches), the image features are extracted using multiple 2D convolutional and pooling layers. These features are transferred to the decoder branch using the skip connection. Multiple skip connections at different encoder layers transfer the layer feature maps to the corresponding decoder layer. The decoder contains multiple decoding convolutional layers and up-sampling layers, and concatenates the transferred encoding feature maps with the up-sampled feature maps. The last layer of the decoder contains a softmax activation function, which generates the microstructure recognition on γ′ phase. Finally, the categorical cross entropy is adopted as the loss function. A detailed overview of the parameters in the network is also presented in Figure 2.

## 4. Experimental Results

To show the superiority of deep transfer learning, we conduct two kinds of transfer learning experiments including transfer from W-900 to W-1000 and transfer from W-1000 to W-900. Wherein each experiment contains different numbers of labeled images from the target domain and full labeled images from the source domain. For comparison, direct training with random initialization is also performed on W-900 and W-1000, respectively.

### 4.1. Evaluation Metric

In this paper, three commonly used metrics including Dice-coefficient, Accuracy, and Intersection over Union (IoU) are adopted for evaluation.

Dice-coefficient is a statistic used to gauge the similarity of two sets, which is defined as follows: (1)Dice(T,P)=2|T∩P|/(|T|+|P|),
where *T* means the labeled γ′ phase and *P* is the predicted γ′ phase.

Accuracy represents the ratio of each positive pixel and negative pixel to be correctly classified. The definition is given as follows: (2)Accuracy(T,P)=(TP+TN)/(|T|+|P|),
where TP means the true positive and TN means the true negative.

IoU is defined as the size of the intersection divided by the size of the union of two sets, which is provided as follows: (3)IoU(T,P)=|T∩P|/(|T|∪|P|).

### 4.2. Recognition Performance

To quantify the performance, the targeted dataset was randomly split into train set, validation set, and test set (60%, 20%, and 20%), respectively. In each experiment, we randomly select *n* (*n* chooses 3, 4, 5, 6, 7, 8) labeled images from the train set to train the classifier, and the best-performing classifier is determined by minimizing the corresponding soft dice loss on the validation set [25]. To eliminate the randomness, we repeat 20 times for each experiment. The average Dice-coefficient, Accuracy, and IoU on the test set are reported in Table 2. We can see that the Dice-coefficient of deep transfer learning outperforms random initialization significantly, especially when *n* is smaller. This phenomenon is consistent with a previous conclusion [26]. It should be noted that deep transfer learning with n=3 can achieve high recognition performance during the transfer from W-900 to W-1000, where the average Dice-coefficient is significantly improved by about 30% compared with random initialization. Meanwhile, deep transfer learning with n=5 can achieve high recognition performance during the transfer from W-1000 to W-900 and has an improvement up to 7% on Dice-coefficient than the random initialization.

To have a visual comparison, Figure 3a,b present the box plots of the Dice-coefficient of two methods at different sample numbers, which is able to show the distribution of these 20 times experimental results comprehensively. In these figures, we can clearly observe the statistical characteristics of experimental results, including upper extreme, upper quartile, median, lower quartile, and lower extreme. It can be seen that as the number of labeled images *n* increases, the Dice-coefficient of random initialization has a more rapid improvement than that of deep transfer learning. It also supports that deep transfer learning can achieve excellent performance when there are only a few labeled images. Meanwhile, we can see that the variance of deep transfer learning is generally smaller than that of random initialization, especially during the transfer from W-900 to W-1000. It means that the deep transfer learning is more robust than the random initialization. Figure 3c visually presents the microstructure recognition performance on γ′ phase. Wherein the green part represents the TP, the red part represents the false negative (FN), the blue part represents the false positive (FP) and the black part represents the TN. We can see that random initialization tends to produce FN when *n* is small. In contrast, deep transfer learning can generally produce TP with a little pixel-level FP and FN. It may owe to the pre-trained process in the deep transfer learning.

### 4.3. Convergence Speed

Note that another advantage of deep transfer learning is the fast convergence [27]. We compare the validation accuracy vs. different epochs of deep transfer learning and random initialization with n=6. The validation accuracy on transfer from W-900 to W-1000 and transfer from W-1000 to W-900 are shown in Figure 3d and e, respectively. We can see that deep transfer learning not only can achieve a classifier with good performance quickly, but is also superior to the random initialization on validation accuracy. To be specific, five epochs are enough for deep transfer learning to obtain a classifier with satisfying generalization ability. In contrast, random initialization needs 10 more epochs for convergence. It means that deep transfer learning helps reduce the training time on γ′ phase recognition. This phenomenon is also similar with a previous conclusion [26].

### 4.4. Discussion

This present research is primarily motivated by the fact that the deep-learning-based image analysis suffers from generalization when the data distribution changes. It means that the manual annotation will be required to train a new model when facing a similar dataset. To overcome this problem, the deep transfer learning method is proposed. It aims to take full advantage of the existing labeled dataset and achieve excellent recognition performance with a few labeled images for a similar dataset. As expected, the deep transfer learning method can shorten the training time and achieve satisfying performance with a few labeled samples. In essence, the deep transfer learning method is equivalent to narrowing the function hypothesis space based on the prior knowledge learned on existing labeled dataset [28], which guarantees the accurate and fast recognition on γ′ phase. Although the success of deep transfer learning method is only validated on two Ni-base superalloys produced at 900 ${}^{{}^{\circ}}$C and 1000 ${}^{{}^{\circ}}$C, it also holds great promise that the proposed method can be applied to the γ′ phase recognition of Ni-base superalloys dataset produced at other temperatures such as 800 ${}^{{}^{\circ}}$C or other multicomponent superalloys dataset such as Al-based.

## 5. Software

To facilitate the Ni-based superalloys microstructure recognition, we have developed and designed a simple software by PyQt Designer [29]. As shown in Figure 4, the overall software interface mainly contains the user input (left part) and real time monitoring (right part). The left part includes the common settings during the deep transfer learning experiment, such as the path of the source domain, the path of the target domain, the choice of network architecture of basic model, the choice of optimizer, the initial learning rate, the number of training epochs, and the use of data augmentation. The deep transfer learning model will start training after the button “run” is pressed. In the upper right window, the model loss and accuracy vs. different epochs are visualized and updated in real time. Meanwhile, the progress bar of the whole training is also presented in the lower right window. In general, the developed software can conduct deep transfer learning experiment through simple operation for the user.

## 6. Conclusions

In this paper, we have proposed a deep transfer learning method for microstructure recognition on γ′ phase, which could use fewer labeled samples to achieve satisfying performance for a new Ni-based superalloys dataset. To evaluate the effectiveness of the proposed method, we have produced and annotated two Ni-based superalloys datasets at temperatures 900 ${}^{{}^{\circ}}$C and 1000 ${}^{{}^{\circ}}$C, respectively. The experimental results have showed that the proposed method only needs five (or less) labeled images to achieve state-of-the-art segmentation accuracy while enjoying the advantage of fast convergence. Finally, we have developed a simple and effective software for the Ni-based superalloys microstructure recognition on γ′ phase, which can facilitate the process of materials characterization and analysis.

Despite the good results, the proposed method still has several shortcomings that can be improved, such as improving the recognition performance or using fewer labeled images. Thus, several open problems deserve further research along the line of the present work. For example, inspired by the success of semi-supervised learning [30,31], how to utilize the unlabeled images in the target domain is a possible solution to improve the recognition performance. Similarly, inspired by the image-to-image translation [32], how to generate the annotated dataset using an existing dataset is a potential solution for unsupervised microstructure recognition. We are currently researching these problems.

## Figures and Tables

**Figure 1 materials-15-04251-f001:**
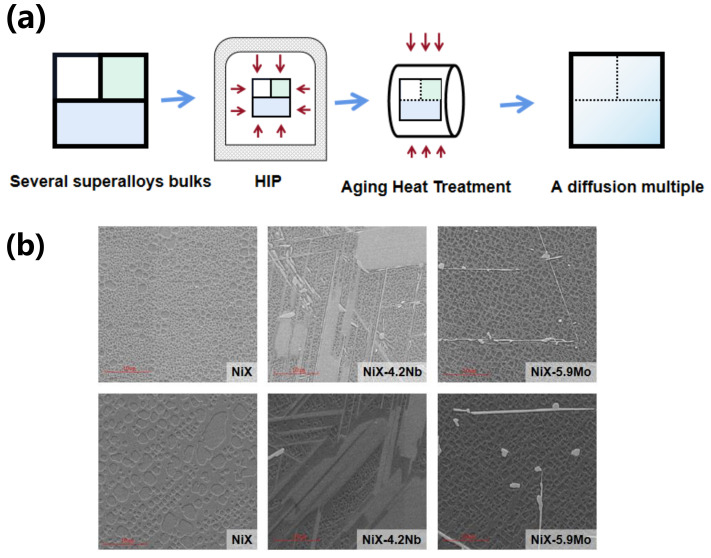
(**a**) The high throughput experimental preparation process of Ni-based superalloys; (**b**) The acquired microstructure by SEM at temperatures 900 ${}^{{}^{\circ}}$C and 1000 ${}^{{}^{\circ}}$C.

**Figure 2 materials-15-04251-f002:**
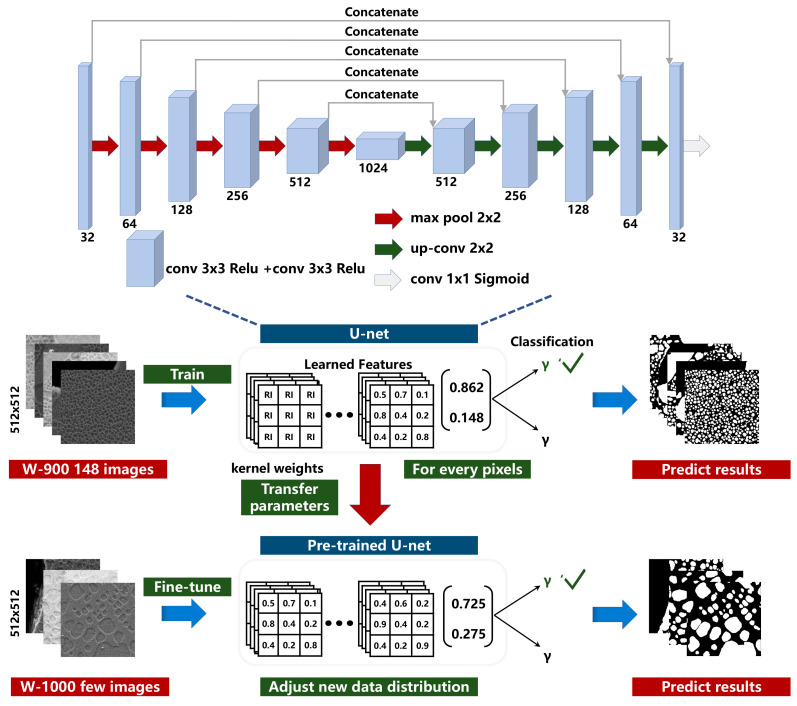
The schematic flowchart of deep transfer learning method from W-900 to W-1000, wherein the U-Net architecture is employed as the basic network.

**Figure 3 materials-15-04251-f003:**
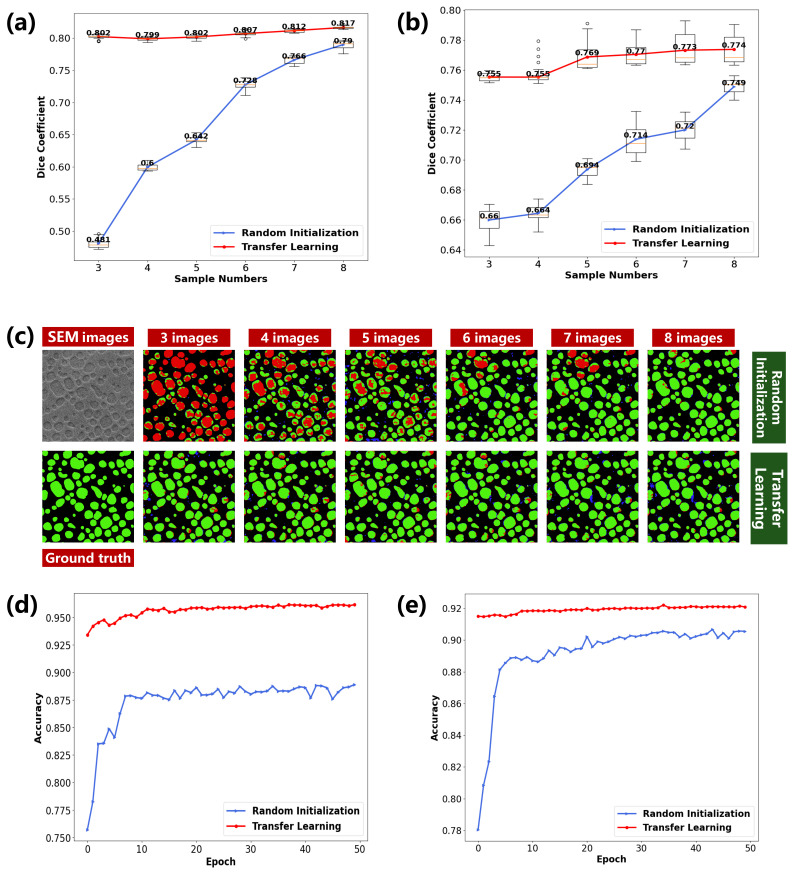
(**a**) The box plot of Dice-coefficient of two methods during the transfer from W-900 to W-1000; (**b**) The box plot during the transfer from W-1000 to W-900; (**c**) Visualization of recognition results; (**d**) The validation accuracy vs. different epochs of two methods during the transfer from W-900 to W-1000; (**e**) The validation accuracy during the transfer from W-1000 to W-900.

**Figure 4 materials-15-04251-f004:**
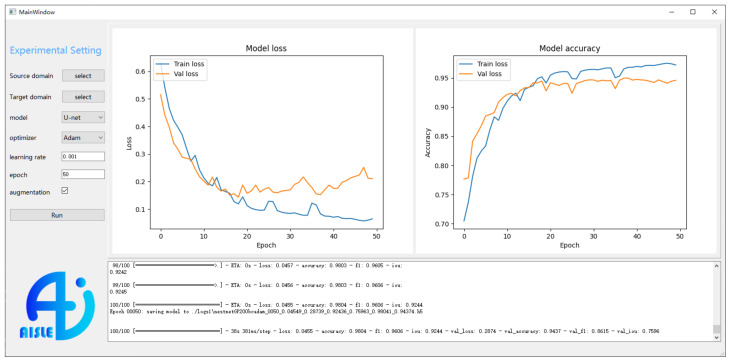
Software interface of deep transfer learning.

**Table 1 materials-15-04251-t001:** Measured composition of simple W1–W7 in wt.%.

	Co	W	Mo	Cr	Al	Ti	Ta	Nb	Hf	Ni
W1	13.0	3.0	2.9	12.2	3.03	3.96	3.1	-	0.21	Bal.
W2	27.8	3.0	2.9	12.0	3.05	4.06	3.0	-	0.21	Bal.
W3	13.0	6.1	2.9	11.8	3.05	3.92	3.0	-	0.22	Bal.
W4	13.1	3.0	6.0	12.1	3.05	4.08	3.1	-	0.19	Bal.
W5	13.0	2.9	3.0	12.0	3.07	6.01	3.0	-	0.19	Bal.
W6	13.0	3.0	2.9	12.0	3.10	4.04	8.1	-	0.20	Bal.
W7	13.0	3.0	3.0	11.9	2.98	4.12	3.0	4.0	0.22	Bal.

**Table 2 materials-15-04251-t002:** Average results of deep transfer learning (TL) and random initialization (RI).

Samples	Method	W-900 to W-1000			W-1000 to W-900		
		Accuracy	Dice	IoU	Accuracy	Dice	IoU
3	RI	80.56%	49.48%	36.46%	80.50%	66.00%	52.34%
	TL	92.03%	80.68%	69.05%	91.76%	75.53%	62.73%
4	RI	83.05%	60.79%	45.46%	82.70%	66.44%	52.29%
	TL	92.00%	80.29%	68.64%	91.87%	75.53%	62.97%
5	RI	89.06%	72.96%	61.02%	82.07%	69.37%	56.71%
	TL	92.09%	80.68%	69.24%	91.97%	76.87%	64.74%
6	RI	87.85%	73.88%	60.84%	86.31%	71.39%	58.76%
	TL	92.25%	81.37%	70.18%	91.66%	77.05%	65.07%
7	RI	87.37%	73.28%	60.28%	87.81%	72.01%	59.73%
	TL	92.15%	81.37%	70.26%	91.90%	77.32%	65.36%
8	RI	90.57%	79.69%	68.61%	89.19%	74.89%	63.63%
	TL	92.15%	81.98%	70.78%	91.93%	77.38%	65.33%

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
