# Peer review of "Deep Transfer Learning for Ni-Based Superalloys Microstructure Recognition on γ′ Phase"

_materials, 2022, doi:10.3390/ma15124251_

Round 1

Reviewer 1 Report

The manuscript "1726484 - Deep Transfer Learning for Ni-based superalloys Microstructure Recognition on γ′ Phase" presents interesting results about the synthesis of new materials. The paper presents relevant information. It is well designed and easy to follow. The applied methodology is adequate and the literature cited is actualized. It deserves publication in Materials after Minor Revisions. The authors must consider the following comments:

  • The abstract contains a brief introduction and the proposal of the article, but does not report the results and main conclusions. It must be completely rewritten.
  • Do not use the words in the title as your keyword. Change it for more innovative words.
  • Improve academic English throughout the manuscript.

Reviewer 2 Report

Dear Authors,

the work is fun and valuable as it saves labor costs and time in microstructure analysis. The quality of the presentation is good, it can be useful for professionals in this field. I would recommend publishing it as is.

Author Response

We are highly grateful to you positive comments for recognizing our contributions. 

Reviewer 3 Report

-        In the Introduction section, the authors cited the specific results of previous research and cited them adequately. However, they did not mention their shortcomings in previous research. In the Introduction section, the penultimate paragraph should contain common features of previous research. The shortcomings of previous research should also be pointed out, in general.

-        In the Introduction section, the last paragraph should contain the scientific contribution and scientific hypotheses of your research. Complete, further elaborate the scientific contribution and scientific hypotheses of your research. Be explicit. In addition to the goal of the research (which was written), the novelty in the context of the scientific contribution should be pointed out. Scientific contributions should be written based on the shortcomings of previous research in the literature. In this way, the authors will better emphasize novelty and scientific soundness.

-        Analyze and discuss possibilities of practical application.

-        In the conclusions, state the scientific contribution, the shortcomings of your methodology and future research.

-        Generally, the quality of the writing could be improved.

Round 2

Reviewer 3 Report

The presented data are original and interesting. The manuscript has been significantly improved and is suitable for publication in the present Journal.